# TLR4 Overexpression Aggravates Bacterial Lipopolysaccharide-Induced Apoptosis via Excessive Autophagy and NF-κB/MAPK Signaling in Transgenic Mammal Models

**DOI:** 10.3390/cells12131769

**Published:** 2023-07-03

**Authors:** Sutian Wang, Kunli Zhang, Xuting Song, Qiuyan Huang, Sen Lin, Shoulong Deng, Meiyu Qi, Yecheng Yang, Qi Lu, Duowei Zhao, Fanming Meng, Jianhao Li, Zhengxing Lian, Chenglong Luo, Yuchang Yao

**Affiliations:** 1State Key Laboratory of Swine and Poultry Breeding Industry, Guangdong Key Laboratory of Animal Breeding and Nutrition, Institute of Animal Science, Guangdong Academy of Agricultural Sciences, Guangzhou 510640, Chinachenglongluo1981@163.com (C.L.); 2Institute of Animal Health, Guangdong Academy of Agricultural Sciences, Guangdong Provincial Key Laboratory of Livestock Disease Prevention Guangdong Province, Scientific Observation and Experiment Station of Veterinary Drugs and Diagnostic Techniques of Guangdong Province, Ministry of Agriculture and Rural Affairs, Guangzhou 510640, China; 3College of Animal Science and Technology, Northeast Agricultural University, Harbin 150030, China; 4Sericultural & Agri-Food Research Institute, Guangdong Academy of Agricultural Sciences, Guangzhou 510640, China; 5Institute of Laboratory Animal Sciences, Chinese Academy of Medical Sciences and Comparative Medicine Center, Peking Union Medical College, Beijing 100021, China; 6Institute of Animal Husbandry, Heilongjiang Academy of Agricultural Sciences, Harbin 150028, China; 7Beijing Key Laboratory for Animal Genetic Improvement, National Engineering Laboratory for Animal Breeding, Key Laboratory of Animal Genetics and Breeding of the Ministry of Agriculture, College of Animal Science and Technology, China Agricultural University, Beijing 100083, China

**Keywords:** Toll-like receptor 4, transgenic animal model, autophagy, apoptosis, inflammation, oxidative stress

## Abstract

Gram-negative bacterial infections pose a significant threat to public health. Toll-like receptor 4 (TLR4) recognizes bacterial lipopolysaccharide (LPS) and induces innate immune responses, autophagy, and cell death, which have major impacts on the body’s physiological homeostasis. However, the role of TLR4 in bacterial LPS-induced autophagy and apoptosis in large mammals, which are closer to humans than rodents in many physiological characteristics, remains unknown. So far, few reports focus on the relationship between TLR, autophagy, and apoptosis in large mammal levels, and we urgently need more tools to further explore their crosstalk. Here, we generated a TLR4-enriched mammal model (sheep) and found that a high-dose LPS treatment blocked autophagic degradation and caused strong innate immune responses and severe apoptosis in monocytes/macrophages of transgenic offspring. Excessive accumulation of autophagosomes/autolysosomes might contribute to LPS-induced apoptosis in monocytes/macrophages of transgenic animals. Further study demonstrated that inhibiting TLR4 downstream NF-κB or p38 MAPK signaling pathways reversed the LPS-induced autophagy activity and apoptosis. These results indicate that the elevated TLR4 aggravates LPS-induced monocytes/macrophages apoptosis by leading to lysosomal dysfunction and impaired autophagic flux, which is associated with TLR4 downstream NF-κB and MAPK signaling pathways. This study provides a novel TLR4-enriched mammal model to study its potential effects on autophagy activity, inflammation, oxidative stress, and cell death. These findings also enrich the biological functions of TLR4 and provide powerful evidence for bacterial infection.

## 1. Introduction

As a pattern recognition receptor (PRR), Toll-like receptor 4 (TLR4) plays an important role in activating innate immunity via recognizing various pathogen-associated molecular patterns, such as lipopolysaccharide (LPS) and lipoteichoic acid of bacteria, structural proteins of viruses, and mannan of fungus [1]. The activated TLR4 would induce its downstream MyD88-dependent or MyD88-independent signaling pathways. Subsequently, MyD88 recruits IRAKs, TRAF6, and Tak1 to form complexes that activate NF-κB and MAPK signaling, triggering inflammatory responses and oxidative stress [2,3]. In the MyD88-independent signaling pathway, adaptor TRIF recruits TRAF3 to induce IRF3 dimerization and translocation, and then boosts activation of NF-κB/MAPK signaling and production of type I interferon, which is crucial for antimicrobial immune responses [4,5,6]. However, TLR4-mediated innate immune responses have dual functions. On one hand, proper TLR4-triggered immune responses help eradicate pathogenic microorganisms and maintain host homeostasis. Conversely, TLR4 over-activation is closely linked to endotoxic shock, autoimmune disease, and cytokine storm, all of which are detrimental to overall health [7]. The expression level of *TLR4* in a host affects the susceptibility of its immune cells to bacteria, the intensity of the inflammatory response, and the removal of pathogenic microorganisms [8,9]. The innate immunity induced by TLR4 plays a vital role in fighting against pathogen infection [10,11].

The primary biological function of autophagy is to degrade damaged organelles, toxic protein aggregates, and even invading microorganisms, to maintain cellular homeostasis [12]. In the host’s anti-infection process, autophagy plays the role of innate immunity against infection by removing intracellular pathogens through lysosomes, and can induce adaptive immunity through antigen presentation [13]. It has been proven that autophagy helps clear various invading pathogens, such as Streptococcus, *Helicobacter*. *Pylori*, and *Pneumonia* [7]. On the other hand, since autophagy is highly conserved, some pathogens have evolved the ability to escape, or even utilize, autophagy to provide a favorable environment for their survival and reproduction [14]. With in-depth study, many researchers believe that the species of pathogenic microorganisms, infection conditions, and the host’s physiological state affect the function of autophagy [15]. Autophagy also has dual roles during pathogenic infection, facilitating the clearance of pathogens and promoting the survival of pathogens [15]. Investigating the role of autophagy in infection can help us prevent infection more effectively. Recently studies have indicated that autophagy is well-connected with TLR4. TLR4-MyD88-MAPK and TLR4-PI3K-Akt pathways regulate the activity of several autophagy initiation complexes [16,17]. In the current study, we develop a *TLR4* overexpression transgenic sheep to investigate the impact of *TLR4* transgenes on autophagy in large mammals.

Appropriate cell death plays an essential role in maintaining the balance of the body’s internal environment and physiological activities. Apoptosis is a typically programmed cell death closely associated with autophagy [18]. Previous studies have discovered potential links (such as Caspase-8 and p62) between autophagy and apoptosis signaling, which could be simultaneously activated in the same tissues and cells [19,20,21]. Moreover, some apoptosis inducers can also regulate autophagy activity, such as etoposide and ceramide activity [22,23]. The relationship between autophagy and apoptosis is very complicated. In some cases, autophagy contributes to cell survival by inhibiting apoptosis, but, in other cases, autophagy and apoptosis can work together to promote cell death [24]. Additionally, some studies suggest autophagy just triggers the occurrence of apoptosis-related characteristics, but does not directly cause cell death [25]. The molecular mechanism underpinning the regulation of cell death by autophagy remains elusive. To date, few studies have explored the interplay between autophagy and apoptosis in large mammals during bacterial infections. We urgently need more research tools to further explore the crosstalk of autophagy and apoptosis.

It is widely known that TLR4 can be activated by bacterial LPS, which can also induce autophagy and apoptosis [26,27]. This study mainly investigates the relationship between TLR4, autophagy, and apoptosis in sheep. Our previous studies have revealed a crucial role of TLR4 in regulating inflammatory responses and oxidative stress, which also have major impacts on apoptosis [28,29,30]. We have also found that the overexpression of TLR4 affected the adhesion and phagocytosis capacity of monocytes/macrophages and the clearance of pathogenic bacteria [31,32]. Nonetheless, the research concerning TLR4 and its regulatory effects on autophagy and apoptosis is scarce. Thus, in this study, we aim to further elucidate the role of TLR4 in modulating autophagy and apoptosis, with particular emphasis on deciphering the molecular mechanism, through which TLR4-mediated autophagy regulates apoptotic signaling. Here, a TLR4-enriched mammal model was generated to investigate specific regulatory mechanisms between autophagy and apoptosis. These findings also enriched the biological functions of TLR4 and provided powerful evidence for bacterial infection.

## 2. Materials and Methods

### 2.1. Animal Ethical Statement

All sample collections and treatments followed the guidelines of the Animal Welfare Committee of the Northeast Agricultural University, and the Animal Welfare Committee approved all experiments of the Northeast Agricultural University (according to the experimental license: NEAU-(2011)-9).

### 2.2. Animals and Cells’ Isolation, Culture, and Identification

The transgenic sheep (German Mutton Merino) was produced through the microinjection of a linearized vector containing *ovis-TLR4* (the *ovis-TLR4* CDS was cloned from the cDNA of sheep kidney) into the pronucleus of fertilized eggs (Figure 1A) [31]. Then, 20 μg genomic DNA was obtained from the ear of the offspring (from 2- to 3-year-old sheep) and was digested by the restriction enzyme *Hind*III (NEB). The probe was designed based on the DNA sequence encompassing the 3′ region of the third exon of TLR4 and the 5′ region of IRES. The detection probe of Southern blotting was amplified with the following primer pair: forward: 5′-ACTGGTAAAGAACTTGGAGGAGG-3′; reverse: ‘5-CCTTCACAGCATTCAACAGACC-3′ (the detection was labelled with digoxigenin), resulting in a 2771-bp exogenous fragment and a 5118-bp endogenous fragment. The monocytes/macrophages of 2- to 3-year-old sheep were isolated from the jugular venous blood with a separation medium (Tbdscience, China). After 2 h of incubation (37 °C, 5% CO_2_), the non-adherent cells were washed out using PBS. The adherent cells were cultivated in RPMI1640 (Gibco, America), containing 10% fetal bovine serum (Gibco, America) (Appendix A). To further confirm these cells, the adherent cells were fixed with 4% paraformaldehyde for 20 min. Then, these cells were blocked and incubated with primary antibody against CD14 (1:500, Bioss, China) or CD11b (1:500, Proteintech, America) at 4 °C overnight, followed by fluorescence-labelled secondary antibody (1:1000, Beyotime, China) at room temperature for 1 h. The cellular nuclei were stained by DAPI. The images were obtained using a fluorescence microscope (Nikon, Japan). The fluorescence was analyzed by ImageJ software (National Institutes of Health; version 1.45). Monocytes/macrophages transfection was achieved using Lipofectamine RNAiMAX (Invitrogen, America) for siRNA-specific TLR4 (Genepharma, China), following the manufacturer’s instructions. The siRNA sequence is shown in Appendix A.

### 2.3. Quantitative Reverse-Transcription Polymerase Chain Reaction (qRT-PCR) Analysis

Total RNA was isolated with TRIzol reagent (Invitrogen, America), and the Prime Script^TM^ RT Reagent kit (TaKaRa, Japan) was used to generate cDNA. The mRNA expression levels of *TLR4*, *IL-1β*, *IL-6*, and *TNF-α* were measured by qRT-PCR on a Roche LightCycler 480 instrument with the SYBR Premix Ex Taq II kit (TaKaRa, Japan). *GAPDH* was chosen as the reference gene. All primer sequences are shown in Appendix A. The 20.0 μL reaction system consisted of 10.0 μL SYBR Premix Ex Taq, 0.8 μL each of the primers (10 μM), 0.4 μL of ROX reference Dye (50×), 2.0 μL of cDNA (200 ng), and 6.0 μL of RNase-free water. The qRT-PCR procedure was as follows: pre-denaturation at 95 °C for 30 s; denaturation at 95 °C for 5 s, 55 °C for 30 s, 72 °C for 30 s for 40 cycles; melt curve at 95 °C for 5 s, 60 °C for 15 s, and 95 °C for 5 s. The results of the mRNA expression were calculated by the 2^−ΔΔCT^ method.

### 2.4. Western Blotting

Cells were harvested and lysed (0 °C, 30 min) in RIPA reagent (Beyotime, China), containing PMSF and a protease and phosphatase inhibitor cocktail (Roche, Switzerland). The protein concentration of each sample was measured by the BCA Protein Assay kit (Thermo, America). Equal amounts of proteins (30 µg each) were incubated at 95 °C for 10 min in an SDS buffer. Proteins were separated by 12% SDS-PAGE and transferred to a polyvinylidene fluoride membrane (Millipore) after incubation overnight with primary antibodies against TLR4 (1:1000; Affinity, America), LC3B (1:1000; Abcam, America), p62 (1:1000; CST), ATG5 (1:1000; Sigma, Germany), p-p65 (1:1000; CST), p-p38 (1:1000; CST), BAX (1:500; CST), BCL-2 (1:500; HUABIO), Caspase-3 (1:1000; Proteintech, America), Caspase-8 P18 (Santa Cruz, America), and GAPDH (1:5000; Proteintech, America) at 4 °C. The membranes were incubated with a horseradish peroxidase-conjugated secondary antibody (1:2000; Beyotime, China) and were then visualized using chemiluminescence (Thermo, America) and quantified by ImageJ software (National Institutes of Health; version 1.45).

### 2.5. Transmission Electron Microscopy Analysis

Sheep monocytes/macrophages were treated with LPS (Sigma, America, L6529) for 12 h. The cells were washed with PBS and fixed with 2.5% glutaraldehyde for 24 h, and then with 1% osmic acid for 1 h. After dehydration in ethanol, the cells were embedded in epoxy resin and observed by an H-7650 microscope at 100 kV (Hitachi, Japan).

### 2.6. Immunofluorescence Microscopy Analysis

After LPS stimulation, the monocytes/macrophages were washed with PBS and fixed with 4% paraformaldehyde for 20 min. Then, 0.3% Triton X-100 was used for PBMC permeabilization for 10 min. After that, the permeabilized cells were blocked and incubated with primary antibody against LC3B (1:200, Abcam, England) at 4 °C overnight, followed by fluorescence-labelled secondary antibody (Solarbio, China) at room temperature for 1 h. The images were obtained using an LSM 710 confocal microscope (Carl Zeiss Jena, Germany).

### 2.7. TUNEL Assay for Detecting Apoptosis

Following the manufacturer’s protocol, a one-step TUNEL kit (Beyotime, China) was used for TUNEL detection. In short, the monocytes/macrophages were fixed in 4% paraformaldehyde for 30 min and permeabilized in 0.3% Triton X-100 for 5 min, followed by incubating with TUNEL for 1 h at 37 °C in a dark place, and the total nuclei were labelled by DAPI (Beyotime, China). The cells were observed with an LSM 710 confocal microscope.

### 2.8. Measurement of Oxidative Stress

After LPS stimulation, the NO and MDA in cell culture supernatants were examined by NO and MDA assay kits (Njjcbio, China), following the manufacturer’s instructions. The intracellular ROS was detected by the Reactive Oxygen Species assay kit (Beyotime, China).

### 2.9. Statistical Analysis

All experiments were performed no fewer than three times, and the significance of all data was assessed by the univariate analysis of variance followed by the unpaired Student’s *t*-test. A result of *p* < 0.05 was considered statistically significant.

## 3. Results

### 3.1. Generation and Identification of Transgenic Individuals Overexpressing TLR4

The *ovis*-*TLR4* CDS were amplified and cloned into the expression vector (pTLR4-3s, Figure 1A). The transgenic offspring were screened by Southern blotting. The transgenic sheep (German Mutton Merino) were produced through microinjection of the linearized *ovis-TLR4* overexpression vector. According to the sequence of the sheep genome and *TLR4* expression vector, the restriction enzyme, *HindIII*, resulted in a 2771-bp exogenous fragment and a 5118-bp endogenous fragment, which both exist in positive transgenic individuals (the wild-type sheep only have a 5118-bp endogenous fragment) (Figure 1B). The relative *TLR4* expression levels of the muscle, liver, kidney, spleen, and stomach of transgenic sheep were analyzed by qRT-PCR. The results showed that TLR4 expression in these organs, except for the stomach, was significantly increased compared to the wild-type sheep (Figure 1D). The growth and weight parameters of these sheep were collected every month. As shown in Figure 1E–H, there were no significant differences in weight, length, height, and chest circumference between wild-type sheep and positive transgenic individuals.

### 3.2. Generation and Identification of Transgenic Individuals Overexpressing TLR4

Further investigation showed that the mRNA and protein levels of TLR4 were significantly higher in transgenic sheep monocytes/macrophages than in wild-type monocytes/macrophages (Figure 2A–C). Generally, activation of TLR4 can activate NF-κB and MAPK signaling pathways. Mechanism analysis suggested that NF-κB and p38 MAPK signaling were involved in many cellular processes. So, we examined these signaling pathways by stimulating monocytes/macrophages with different LPS treatments (0, 1, and 100 μg/mL for 12 h). Stimulating monocytes/macrophages with LPS increased phospho-p65 and phospho-p38, and their levels were significantly higher in the monocytes/macrophages of transgenic sheep (Figure 2D–F). Taken together, the *TLR4* expression level was higher in the immunocytes of the TG group, and LPS stimulation could more efficiently activate NF-κB and MAPK signaling in monocytes/macrophages of transgenic individuals.

### 3.3. Overexpression of TLR4 Promotes the Production of Pro-Inflammatory Cytokines and Oxidative Stress

Since LPS-mediated production of pro-inflammatory cytokines and oxidative stress was related to TLR4-MyD88-dependent NF-κB or MAPK activation, we next determined if the pro-inflammatory cytokines and oxidative stress levels between the TG and WT groups were any different. All IL-1β, IL-6, and TNF-α mRNA expression levels in transgenic sheep are markedly higher than those in the wild-type group with high doses of LPS treatment (100 μg/mL) (Figure 3A–C). Similarly, the overexpression of TLR4 also dramatically promotes ROS, NO, and MDA productions (Figure 3D,E). More notably, as the intensity of the LPS treatment increased, the production of pro-inflammatory cytokines and oxidative stress also increased. These results suggest that the overexpression of TLR4 aggravates inflammation and oxidative stress under high-dose LPS treatment. These results indicate that TLR4 is crucial in regulating LPS-induced pro-inflammatory cytokines and oxidative stress.

### 3.4. Overexpression of TLR4 Leads to Dysfunctional Autophagy

When animals’ bodies and cells sense external stress, they undergo a series of physiological changes in response to these stimuli, including autophagy. Autophagy is a conserved defense mechanism that plays various functions under different physiological pressures. The formation of autophagosomes/autolysosomes is one of the crucial processes of intracellular autophagy. Monocytes/macrophages from each group were detected by transmission electron microscopy (TEM, Figure 4) to examine autophagosomes in sheep. As the dose of LPS increased, the number of autophagosomes/autolysosomes rose rapidly. The number of autolysosomes in the TG group was significantly higher than in the WT group. Furthermore, the cell structure was incomplete and displayed numerous autophagic vacuolization in the TG LPS++ group (LPS, 100 μg/mL). We used laser scanning confocal microscopy to further detect the autophagy activity by examining autophagy-related protein LC3B level. The nuclei were tagged by hoechst33342 (blue), and LC3B protein was tagged by Cy3 (red). There is a distinct increase of LC3B in the TG group compared to the WT group (Figure 5A). During autophagy initiation, LC3B-I is converted to LC3B-II. The ratio of LC3B-II/GAPDH in transgenic groups was significantly higher than in the wild-type group, indicating increased autophagy activation in transgenic sheep (Figure 5B,C). The ATG5 protein level in TG groups was also higher than in the WT group, suggesting TLR4-overexpression promotes the formation of autophagosomes (Figure 5B,D). Interestingly, the expression of p62 was significantly lower in transgenic sheep after 1 μg/mL LPS stimulation in monocytes/macrophages, but the expression of p62 was significantly higher in transgenic sheep after 100 μg/mL LPS stimulation in monocytes/macrophages (Figure 5B,D). p62 is a receptor protein related to the delivery of cargo. The decreased p62 protein and the increased LC3B-II/GAPDH ratio indicate the promoted autophagic flux. The increase of p62, accompanied by the rise of the LC3B-II/GAPDH ratio, indicates the blockage of autophagic flux. These results suggest that a high dose of LPS (100 μg/mL) may block autophagic degradation in monocytes/macrophages of transgenic sheep. This result also corresponds to the presence of large amounts of autophagosomes/autolysosomes in the TG LPS++ group in Figure 4A.

### 3.5. TLR4 Overexpression Promotes LPS-Induced Apoptosis

Some studies have found that LPS stimulation, activation of NF-κB signaling, and dysfunctional autophagy could cause apoptosis. The results of TEM suggest that apoptosis may have occurred. So, we next determine the levels of apoptosis-related markers. Firstly, we treat monocytes/macrophages with LPS (LPS+, 1 μg/mL, and LPS++, 100 μg/mL) in TLR4-overexpressing sheep and wild-type sheep and examine cell apoptosis using TUNEL. As the LPS dosage increases, the ratios of TUNEL-positive apoptotic monocytes/macrophages also increase. Furthermore, the percentages of TUNEL-positive cells in the TG group are significantly higher than that of the WT group under a high concentration of LPS treatment (Figure 6A,B). Additionally, we also detected the protein levels of proapoptotic BAX, antiapoptotic BCL-2, and apoptosis-related protein Caspase-3 and Caspase-8. The results showed an increasing level of BAX and cleaved Caspase-3 and cleaved Caspase-8 in the TG group compared to the WT group. The BCL-2 level in the TG group was lower than that of the WT group (Figure 6C). Analysis of the BAX/BCL-2 ratio showed that the BAX/BCL-2 ratio rises dramatically in the TG group compared to the WT group. The high BAX/BCL-2 ratio and cleaved Caspase-3 and cleaved Caspase-8 indicated there were a great number of apoptotic cells. Taken together, LPS stimulation aggravates apoptosis in TLR4-overexpressing transgenic individuals.

### 3.6. Knockdown of TLR4 Affects Autophagy and the Production of Pro-Inflammatory Cytokines and Oxidative Stress

To further investigate the function of TLR4 during autophagy, inflammation, and oxidative stress, RNAi was used to knock down TLR4. After transfecting si-TLR4, the protein level of TLR4 significantly decreased (Figure 7A,B), and inhibition of TLR4 down-regulated the activation of TLR4-downstream NF-κB and p38 MAPK signals (Figure 7A,C,D). Moreover, the inhibition of TLR4 decreased the ATG5 levels and LC3B-II/GAPDH ratio, which indicated that TLR4 participated in the formation of autophagosomes (Figure 7A,E,F). In addition, the knockdown of TLR4 also promoted p62 degradation, which may alleviate the blockage of autophagic flux. On the other hand, TLR4 inhibition repressed the expression of IL-6, TNF-α, and the production of NO, MDA, and cellular ROS (Figure 7H–L). These results showed that the knockdown of TLR4 decreased inflammatory responses and oxidative stress and suppressed excessive autophagy.

### 3.7. Inhibition of TLR4 and Autophagy Alleviates Apoptosis

To explore the relationship between TLR4, autophagy, and apoptosis in monocytes/macrophages, we used the si-TLR4, autophagy inhibitor 3-methyladenine (3MA), and lysosomal protease inhibitor E64d (Figure 8). TLR4 inhibition and 3MA significantly alleviated LPS-induced apoptosis, while E64d did not suppress LPS-induced apoptosis (Figure 8A,B). Furthermore, we also examine the apoptosis-related protein level. TLR4 inhibition and 3MA decreased BAX/BCL-2 ratio and the level of cleaved Caspase-3, while E64d increased BAX/BCL-2 ratio and Caspase-3 activity (not significant vs. LPS group) (Figure 8C–E). These results indicate that the elevated TLR4 may be related to the accumulation of autophagosomes/autolysosomes, which is associated with LPS-induced apoptosis. LPS-induced accumulation of autophagosomes/autolysosomes may contribute to sheep monocytes/macrophages apoptosis.

### 3.8. NF-κB, p38 MAPK Signaling Pathways, and ROS Are Involved in Autophagy-Mediated Cell Death

Although the above results revealed a close relationship between TLR4, autophagy, and apoptosis, we still wanted to explore which signals are involved in this process. After TLR4 recognizes LPS stimulation, TLR4-downstream NF-κB and p38 MAPK signaling pathways are activated. These signals promote the production of pro-inflammatory cytokines, oxidative stress, and autophagy. Additionally, excessive inflammation, autophagy and oxidative stress contribute to apoptosis. So, we speculate that NF-κB and p38 MAPK signaling may be associated with TLR4-regulated apoptosis. We used NF-κB inhibitor BMS-345541, p38 MAPK-specific inhibitor SB203580, and free radical scavenger NAC to test this hypothesis (Figure 9). After treating cells with these inhibitors, we detected autophagy and apoptosis-related protein levels. Western blotting analysis showed that the inhibited NF-κB, p38 MAPK signaling, and oxidative stress decreased protein levels of ATG5, LC3B-II/GAPDH ratio, BAX/BCL-2 ratio, and cleaved Caspase-3 (Figure 9A–F). Also, suppressing NF-κB, p38 MAPK signaling, and oxidative stress decreased the ratios of TUNEL-positive cells, which indicated that the monocytes/macrophages apoptosis was alleviated (Figure 9G). Moreover, the decreasing NF-κB activity downregulated the expression of inflammatory cytokine (Figure 9H–J). These results suggest that NF-κB, p38 MAPK signaling, oxidative stress, and inflammation participated in autophagy-mediated cell death.

## 4. Discussion

Bacterial infections badly impact livestock production, the social economy, and public health [7]. *E. coli* and *Brucella*, for instance, are the most concerning gram-negative bacterium in the sheep/goat industry [33,34]. As one of the important PRRs, TLR4 recognizes bacterial LPS and induces the host’s innate immune responses, promoting innate immune system resistance to infections [1]. TLR4-mediated MyD88-dependent/independent signals leads to nitroxidative stress, inflammation, and type I IFN response, which participates in the clearance of pathogens [35,36]. Meanwhile, the TLR4 expression level is closely related to pathogen infection. TLR4-deficient mice were hyporesponsive to LPS stimulation [8,37]. TLR4 deficiency could promote bacteria proliferation in mutant mice [38]. In contrast, overexpressing TLR4 seemed to resist gram-negative bacterial infections [39]. In our study, the TLR4 expression in transgenic sheep is significantly higher than that of control sheep. I overexpression of TLR4 could enhance the activations of NF-κB and p38 MAPK with increasing doses of LPS (Figure 2). In the sterilization process, the activations of NF-κB and MAPK signals are always entangled with oxidative stress and inflammatory responses [40]. As expected, TLR4 overexpressing increased LPS-induced production of pro-inflammatory cytokines and free-radical intermediates with increasing doses of LPS (Figure 3).

It is widely acknowledged that autophagy is a conserved physiological, compensatory metabolic process that helps cells maintain homeostasis by moderately degrading toxic proteins and damaged organelles [41]. However, more and more studies have shown that excessive autophagy harms the organism [42,43]. Currently, activation and regulation of autophagy and multiple functions of autophagy in infections are research focuses. With in-depth research, many studies suggest that TLR4 contributes to autophagy induction and regulation during different physiological stresses [44,45,46]. On the other hand, TLR4 activation promotes the production of proinflammatory cytokines and nitroxidation intermediates via the NOX signaling pathway, which, in turn, activates Nrf2 and AMPK pathways. This ultimately leads to the transcriptional upregulation of key autophagy-related genes, such as NIX and BNIP3, thereby regulating autophagy activity [7]. In this research, the elevated TLR4 promoted autophagy initiation by raising the ratio of LC3B-II/GAPDH and upregulating ATG expression levels. More autophagosomes/autolysosomes and LC3 dots suggested that TLR4 overexpression could enhance autophagy activity by promoting autophagy initiation (Figure 4 and Figure 5). We further use Baf A1 to evaluate autophagic flux. Indeed, the Baf A1 treatment experiment confirmed that TLR4 overexpression increased the synthesis of autophagosome (Appendix A). Generally, the increased autophagy activity may help remove toxic proteins and invading pathogens [47,48]. Surprisingly, with a high dose of LPS (100 μg/mL) stimulation, the structure of transgenic monocytes/macrophages became incomplete and displayed numerous vacuoles, which may be a kind of “autophagy-mediated cell death”. So, we continued to explore whether the autophagic flux was impaired. Our further research showed the p62 expression in the TG group was upregulated with the high-dose LPS stimulation, which indicated that the autophagic flux was impaired (Figure 5). Typically, an increased autophagy activity depends on a decreased expression of p62. However, an increased p62 expression might suggest that autophagic degradation was blocked [44,49]. It is worth noting that TLR4 has different regulation effects on autophagy activity in different studies [40,50,51]. The observed variations in outcomes could be accounted for by discrepancies in subjects, physiological states, and stimulus conditions. Of greater significance, however, is the fact that TLR4-mediated oxidative stress, innate immune responses, and autophagy exhibit a dual role in combating pathogenic infections. In other words, controlled levels of oxidative stress, inflammatory responses, and autophagy are beneficial in pathogen elimination, whereas excessive amounts can lead to detrimental effects on host tissues and organs [52,53,54]. Our results showed that knocking down *ovis-TLR4* could effectively suppress NF-κB and p38 MAPK signal transduction, downregulate autophagy activity, and decrease the production of oxidative intermediates and proinflammatory cytokines (Figure 7). Because of the high-dose LPS challenge, inhibition of TLR4 suppressed autophagy initiation by downregulating the ATG5 protein level, which played a crucial role in the formation of autophagosomes [55]. Furthermore, inhibiting TLR4 expression appears to alleviate the high-dose LPS-induced blockage of autophagic degradation by increasing autophagic flux (Figure 7). We first used a large transgenic animal model to reveal that TLR4 was involved in LPS-induced autophagy initiation and affects autophagic degradation.

Although early studies identified autophagy as a mechanism for cell survival, increasing evidence suggested that autophagy also regulates the process of cell death [56,57]. Despite this, we did not observe the typical characteristics of apoptosis, such as cell shrinkage and karyorrhexis, activation of NF-κB and MAPK signaling, and dysfunctional autophagy always triggered apoptosis [58,59,60]. It is of great significance for the apoptosis of macrophages to fight against invading pathogens [61]. Many studies have indicated that LPS-induced apoptosis of macrophages is associated with the TLR4 signaling pathway [62,63]. Additionally, many proinflammatory cytokines, such as TNF-α and IL-1β, and oxidative intermediates could trigger apoptosis [64,65,66]. Recently, more and more evidence show that autophagy and apoptosis can coordinate, antagonize, or promote each other, affecting cell fate [24,67]. Based on these findings and perspectives, we evaluated the effect of overexpression of TLR4 on the apoptosis of sheep monocytes/macrophages. By detecting apoptosis markers and DNA damage, we found that overexpressing TLR4 could promote LPS-induced apoptosis of monocytes/macrophages. The variation trend of monocytes/macrophages apoptosis rate was in agreement with the LPS dose and the expression level of *TLR4*. Most noteworthy, high-dose LPS stimulation induced severe apoptosis in transgenic individuals (the TUNEL-positive rate was up to 60%). Similarly, knocking down TLR4 decreased the DNA damage level and proapoptotic BAX protein expression, with increasing anti-apoptotic BCL2 protein expression, which alleviated apoptosis (Figure 8). Moreover, we also knocked down ATG5 to block autophagy. The results showed that LC3B-II/GAPDH ratio decreased upon ATG5 knockdown. Meanwhile, ATG5 inhibition decreased the BAX/BCL-2 ratio and the levels of cleaved Caspase-3 and Caspase-8, which suggested that inhibition of autophagy alleviated LPS-induced apoptosis in the TLR4-overexpressed group (Appendix A). Some studies have suggested that elevated autophagy levels could induce cell death by activating apoptosis under extreme conditions (overexpression of crucial autophagy-initiation-related proteins) [68,69]. Our results suggested the apoptosis level of sheep monocytes/macrophages is closely linked to the autophagy activity and TLR4 expression under high-dose LPS stress.

A complete autophagy process consists of three crucial stages: autophagosome formation stage, autophagosome-lysosome fusion stage, and autolysosome degradation stage [70,71]. In the process of autophagy, if autophagy is blocked at different stages, cells may fail to adapt to the stress, leading to apoptosis [18,72]. Therefore, we were eager to determine whether the transgenic PBMC apoptosis was caused by dysfunctional autophagy under LPS stress. According to the above results, we speculated that the excessive accumulation of autophagosomes/autolysosomes might be the leading cause of apoptosis. Therefore, we selected autophagosome formation inhibitor 3MA and autolysosome degradation inhibitor E46d to explore the relationship between autophagy and apoptosis. Our study showed that 3MA could attenuate LPS-induced apoptosis, whereas E64d treatment appears to aggravate apoptosis (but the difference was not statistically significant). So, we hypothesized that the excessive accumulation of autophagosomes/autolysosomes might contribute to LPS-induced apoptosis in transgenic monocytes/macrophages. Although many researchers believe that autophagy helps alleviate apoptosis by timely degradation of damaged mitochondria and pre-apoptotic proteins, other studies have indicated that high-level autophagy could promote apoptosis by activating Caspase signaling or regulating intracellular fatty acid metabolism [21,69,73]. It has been realized that autophagy-mediated apoptosis depends on the cell types, stress levels, and physiological state of the organism [25,74]. In addition, it is widely acknowledged that activation of the TLR4-MyD88-NF-κB/MAPK signaling axis can promote the production of various inflammatory cytokines, which are responsible for oxidative stress, autophagy, and apoptosis [75,76,77,78]. In our TLR4-overexpressing animal model, we also found that inhibiting TLR4 downstream NF-κB and p38 MAPK signaling, or suppressing oxidative intermediates production, contributed to downregulating autophagy activity and alleviating apoptosis (Figure 9). When pathogenic microorganisms invade, the host’s inflammatory responses, oxidative stress, autophagy, and apoptosis serve as double-edged swords. This study proposes that low-dose LPS stimulation induces appropriate levels of inflammation, autophagy, and apoptosis to maintain cellular homeostasis. However, high-level LPS stress induces the production of a considerable number of pro-inflammatory cytokines and autolysosomes, ultimately leading to severe apoptosis. These results are mainly attributed to TLR4 overexpression. On the one hand, TLR4 overexpression induced considerable inflammatory cytokines and nitroxidation intermediates via the NF-κB-NOX signal. On the other hand, the TLR4- NF-κB/MAPK signaling axis might affect autophagy activity by targeting p62, Beclin1, mTOR, BNIP3, and NIX [7,12,37]. Moreover, autophagic dysfunction leads to the failure of the elimination of inflammatory cytokines and nitroxidation intermediates, thus exacerbating apoptosis. According to our results, we deduced that the different autophagy activity was attributed to the expression level of *TLR4*. When LPS stimulation was at low levels, TLR4-overexpression helped induce moderate autophagy to remove these harmful molecules and maintain homeostasis. However, under high-level LPS stress, overexpression of TLR4 caused lysosomal dysfunction and impaired autophagic flux. The excessive accumulation of autophagosomes/autolysosomes can further trigger severe apoptosis in transgenic monocytes/macrophages, leading to disruptions in bodily homeostasis. Additionally, dysfunctional autophagy fails to timely degrade the excessive proinflammatory cytokines and oxidative intermediates mediated by TLR4 overexpression, ultimately resulting in tissue and organ damage. These results indicated that TLR4 had dual roles in the process of maintaining homeostasis.

## 5. Conclusions

In summary, our current study used a TLR4-enriched transgenic mammal model to explore the underlying mechanism of autophagy-mediated apoptosis. LPS stress triggers autophagosomes/autolysosomes accumulation and autophagy disorder via the TLR4-dependent signaling pathway, aggravating apoptosis of transgenic monocytes/macrophages. Specifically, the elevated TLR4 aggravates LPS-induced monocytes/macrophages apoptosis by leading to lysosomal dysfunction and impaired autophagic flux, which is associated with TLR4 downstream NF-κB and MAPK signaling pathways. Furthermore, inhibition of TLR4 and autophagy can ameliorate apoptosis in transgenic individuals. This study reveals new mechanisms for further understanding the multiple functions of TLR4 during bacterial invasion. However, we have not yet found the key node where the TLR4 signaling pathway directly interacts with autophagy-related proteins. This work requires further exploration. We have also provided new preventive strategies for bacterial infection. Based on our results, we drew a diagram illustrating the proposed signaling of the effects of TLR4 on autophagy-mediated apoptosis (Figure 10).

## Figures and Tables

**Figure 1 cells-12-01769-f001:**
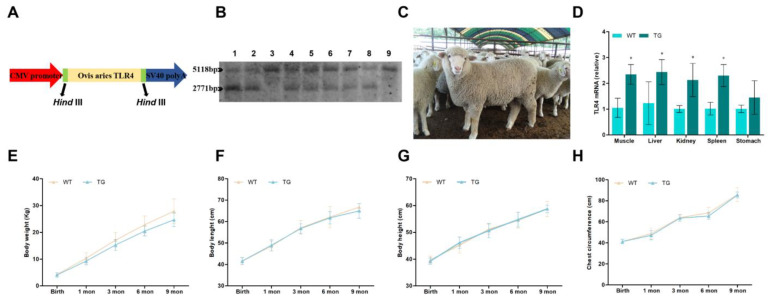
Generation, identification, and general status of the transgenic sheep overexpressing *TLR4*. (**A**) Construction of the *ovis-TLR4* overexpression vector. (**B**) Southern blot analysis for the genome of transgenic animals. The transgenic animals had the endogenous 5118-bp *TLR4* band and the exogenous 2771-bp *TLR4* signature band. The transgenic sheep were: 1, 2, 4, 5, 6, 7, 8, and the wild-type sheep were 3 and 9. (**C**) Photo of the positive transgenic sheep. (**D**) The expression of *TLR4* mRNA in different organs was analyzed by qRT-PCR. (**E**) Body-weight analysis of transgenic sheep at different ages. (**F**) Body-length analysis of transgenic sheep at different ages. (**G**) Body-height analysis of transgenic sheep at different ages. (**H**) Chest circumference analysis of transgenic sheep at different ages. WT, wild-type sheep; TG, transgenic sheep. All data are presented as the mean ± SEM, n ≥ 3; * *p* < 0.05 vs. WT group.

**Figure 2 cells-12-01769-f002:**
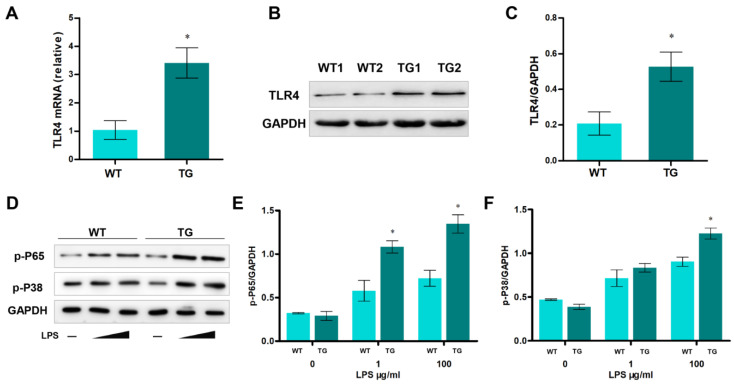
Overexpression of TLR4 leads to upregulating NF-κB and p-38 MAPK signaling in monocytes/macrophages. (**A**) The expression of *TLR4* mRNA was analyzed by qRT-PCR. (**B**,**C**) Western blotting analysis of TLR4 protein level. (**D**–**F**) The p65 and p-P38 proteins’ levels were examined by Western blotting. WT, wild-type sheep; TG, transgenic sheep. All data are presented as the mean ± SEM, n ≥ 3; * *p* < 0.05 vs. WT group.

**Figure 3 cells-12-01769-f003:**
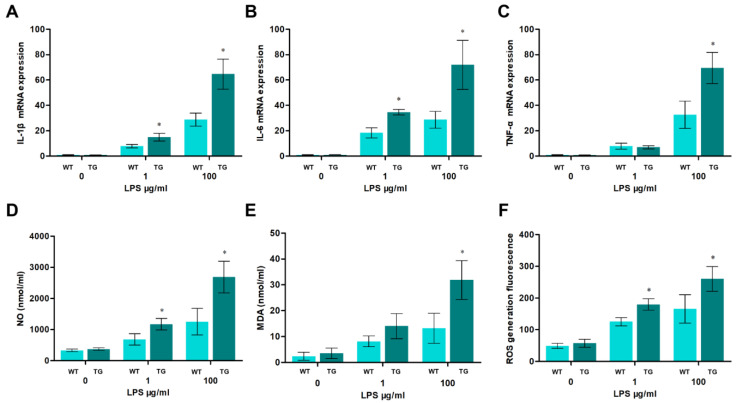
Pro-inflammatory cytokines and oxidative stress levels in sheep monocytes/macrophages under LPS stimulation. (**A**–**C**) The expression of IL-1β, IL-6, and TNF-α mRNA in sheep monocytes/macrophages were examined by qRT-PCR at different LPS treatments (0, 1, and 100 μg/mL for 12 h). (**D**,**E**) The activities of NO and MDA content. (**F**) DCFH-DA assay to determine ROS levels. WT, wild-type sheep; TG, transgenic sheep. All data are presented as the mean ± SEM, n ≥ 3; * *p* < 0.05 vs. WT group.

**Figure 4 cells-12-01769-f004:**
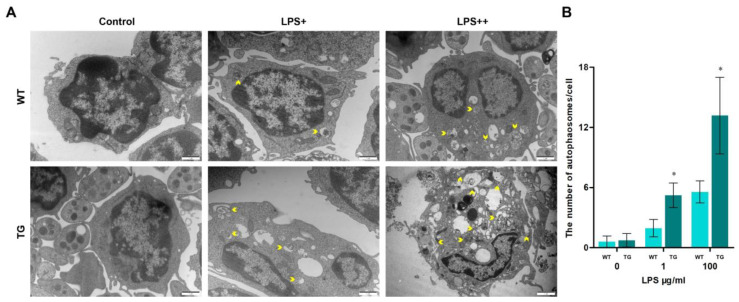
Autophagosomes increase in transgenic sheep. (**A**) Transmission electron microscopy of glutaraldehyde-fixed monocytes/macrophages treated with LPS (LPS+, 1 μg/mL, and LPS++, 100 μg/mL), the yellow arrows highlight autophagosomes/autolysosomes; scale bar: 1 μm. (**B**) Statistical data of the number of autophagosomes/autolysosomes per cell; the graph quantifies no fewer than 30 cells per group. WT, wild-type sheep; TG, transgenic sheep. All data are presented as the mean ± SEM, n ≥ 3; * *p* < 0.05 vs. WT group.

**Figure 5 cells-12-01769-f005:**
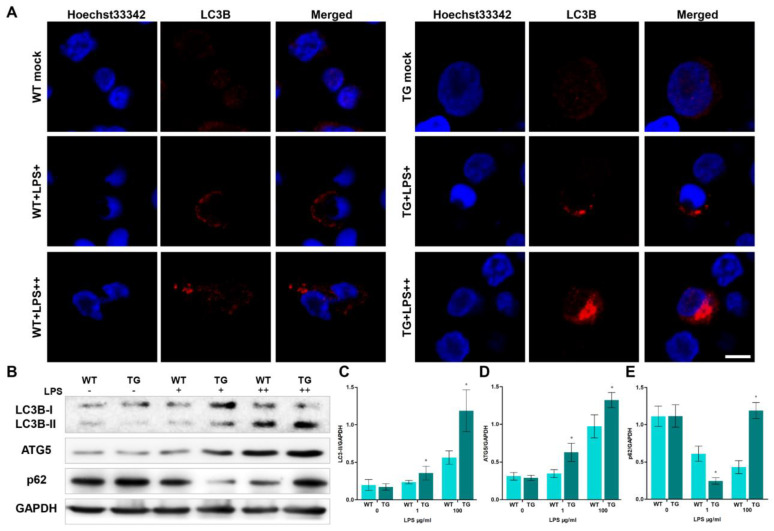
A high dose of LPS treatment induces dysfunctional autophagy of transgenic sheep monocytes/macrophages. (**A**) monocytes/macrophages were treated with LPS (LPS+, 1 μg/mL, and LPS++, 100 μg/mL), and the cells were fixed and stained with antibodies against LC3B. Red: Cy3-labeled LC3B protein; Blue: Hoechst33342-labeled nuclei; Scale bars: 5 μm. (**B**) LC3B, ATG5 and p62 proteins were examined by Western blotting. GAPDH was used as a loading control. (**C**–**E**) LC3B-II/GAPDH, ATG5/GAPDH, and p62/GAPDH ratios of each group, based on immunoblotting. WT, wild-type sheep; TG, transgenic sheep. All data are presented as the mean ± SEM, n ≥ 3; * *p* < 0.05 vs. WT group.

**Figure 6 cells-12-01769-f006:**
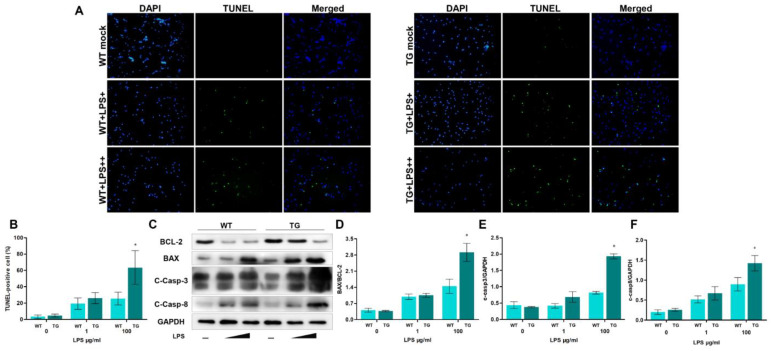
A high dose of LPS stimulation aggravates apoptosis in transgenic sheep. (**A**) monocytes/macrophages were treated without or with LPS (LPS+, 1 μg/mL, and LPS++, 100 μg/mL), and then TUNEL was used to determine the apoptosis. Green: TUNEL-positive nuclei; Blue: DAPI-labeled total nuclei, ×200. (**B**) Statistical data of the ratio of TUNEL-positive apoptotic monocytes/macrophages. (**C**) BCL-2, BAX, cleaved Caspase-3, and cleaved Caspase-8 proteins were examined by Western blotting. GAPDH was used as a loading control. (**D**) BAX/BCL-2 ratios of each group, based on immunoblotting. (**E**) C-Caspase-3/GAPDH ratios of each group, based on immunoblotting. (**F**) C-Caspase-8/GAPDH ratios of each group, based on immunoblotting. WT, wild-type sheep; TG, transgenic sheep. All data are presented as the mean ± SEM, n ≥ 3; * *p* < 0.05 vs. WT group.

**Figure 7 cells-12-01769-f007:**
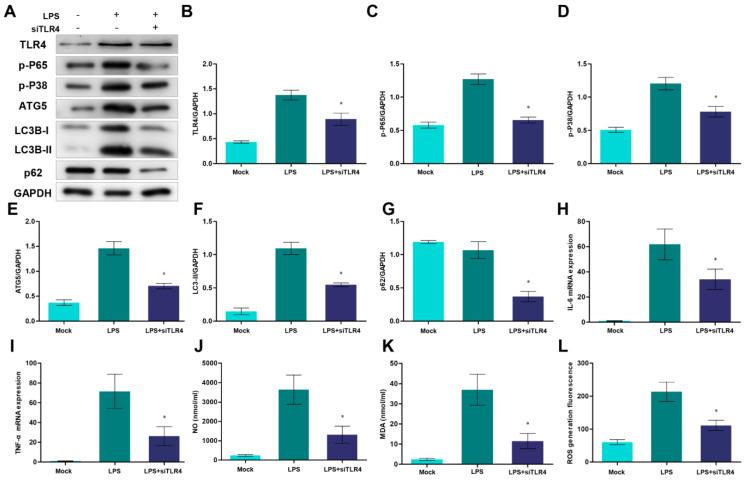
Knockdown of TLR4 decreases autophagosome formation, pro-inflammatory cytokines production, and oxidative stress levels. (**A**) TLR4, p65, p-P38, ATG5, LC3B, and p62 proteins were examined by Western blotting. GAPDH was used as a loading control. (**B**–**G**) TLR4/GAPDH, p65/GAPDH, p-P38/GAPDH, ATG5/GAPDH, LC3B-II/GAPDH, and p62/GAPDH ratios of each group, based on immunoblotting. (**H**,**I**) The expression of IL-6 and TNF-α mRNA in sheep monocytes/macrophages were examined by qRT-PCR. (**J**,**K**) The activities of NO and MDA content. (**L**) DCFH-DA assay to determine ROS levels. All data are presented as the mean ± SEM, n ≥ 3; * *p* < 0.05 vs. LPS group.

**Figure 8 cells-12-01769-f008:**
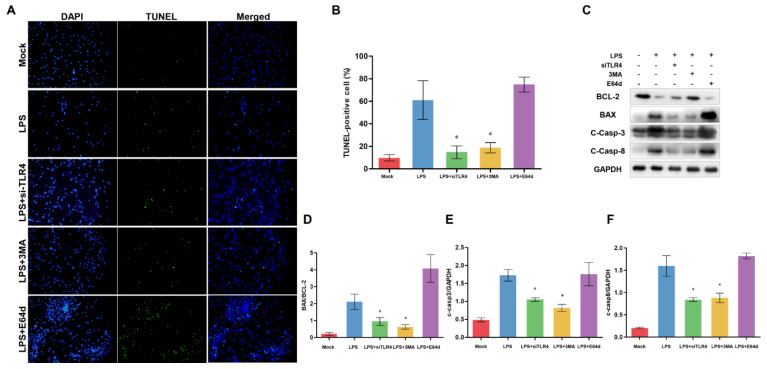
Inhibition of TLR4 and autophagy alleviates apoptosis in TLR4-overexpressing monocytes/macrophages of sheep. (**A**) monocytes/macrophages were treated either with si-TLR4, 3MA, or E64d, and then TUNEL was used to determine the apoptosis. Green: TUNEL-positive nuclei; Blue: DAPI-labeled total nuclei, ×200. (**B**) Statistical data of the ratio of TUNEL-positive apoptotic monocytes/macrophages. (**C**) BCL-2, BAX, cleaved Caspase-3, and cleaved Caspase-8 proteins were examined by Western blotting. GAPDH was used as a loading control. (**D**) BAX/BCL-2 ratios of each group, based on immunoblotting. (**E**) C-Caspase-3/GAPDH ratios of each group, based on immunoblotting. (**F**) C-Caspase-8/GAPDH ratios of each group, based on immunoblotting. All data are presented as the mean ± SEM, n ≥ 3; * *p* < 0.05 vs. LPS group.

**Figure 9 cells-12-01769-f009:**
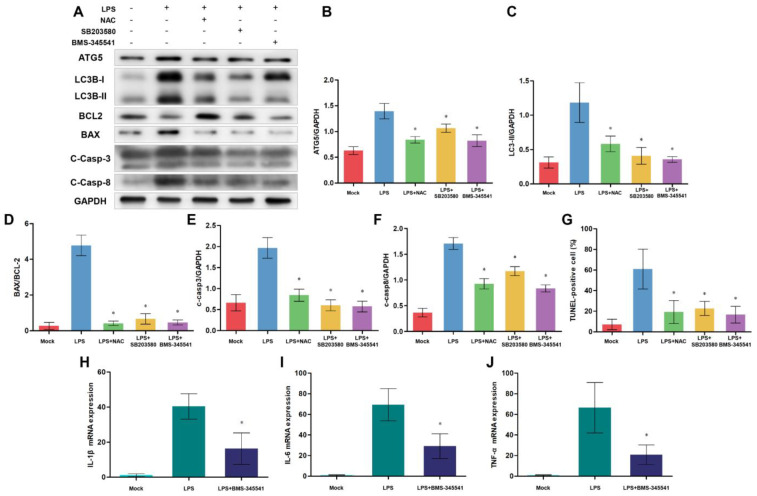
NF-κB and p38 MAPK signaling pathways and ROS effect on autophagy-mediated cell death. (**A**) ATG5, LC3B, BCL-2, BAX, cleaved Caspase-3, and cleaved Caspase-8 proteins were examined by Western blotting. GAPDH was used as a loading control. (**B**–**F**) ATG5/GAPDH, LC3B-II/GAPDH, BAX/BCL-2, C-Caspase-3/GAPDH, and C-Caspase-8/GAPDH ratios of each group, based on immunoblotting. (**G**) Statistical data of the ratio of TUNEL-positive apoptotic monocytes/macrophages. (**H**–**J**) The expression of IL-1β, IL-6, and TNF-α mRNA in sheep monocytes/macrophages were examined by qRT-PCR. All data are presented as the mean ± SEM, n ≥ 3; * *p* < 0.05 vs. LPS group.

**Figure 10 cells-12-01769-f010:**
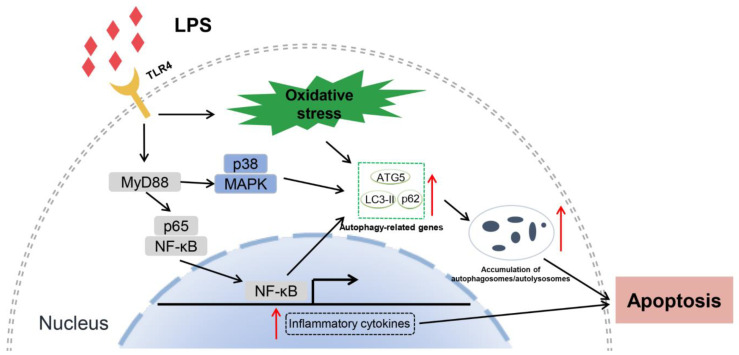
A diagram illustrates the proposed signaling of the effects of TLR4 on autophagy-mediated apoptosis. TLR4-overexpressing-mediated dysfunctional autophagy would cause monocytes/macrophages apoptosis in transgenic animals by accumulating autophagosomes/autolysosomes. The dysfunctional autophagy was associated with TLR4 downstream NF-κB and p38 MAPK signaling and oxidative stress.

## Data Availability

Not applicable.

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
