# Peer review of "TLR4 Overexpression Aggravates Bacterial Lipopolysaccharide-Induced Apoptosis via Excessive Autophagy and NF-κB/MAPK Signaling in Transgenic Mammal Models"

_cells, 2023, doi:10.3390/cells12131769_

Round 1
Reviewer 1 Report (Previous Reviewer 1)
1. Authors do not claim how they confirm monocytes/macrophage cells. Please provide a methodological description for confirmation of monocytes/macrophages and provide the results in a supplementary data sheet.
Author Response
Comments and Suggestions for Authors: Authors do not claim how they confirm monocytes/macrophage cells. Please provide a methodological description for confirmation of monocytes/macrophages and provide the results in a supplementary data sheet.
Response: We appreciate your professional advice and pay our homage to your kind work. According to your suggestion, we provide the methodological description for confirmation of monocytes/macrophages (see section 2.2 Animals and cells isolation, culture and identification). We also provide the results in the supplemental file. All the changes in the revised manuscript are highlighted in the “Track Changes”. Please check the submitted revision manuscript.
Reviewer 2 Report (New Reviewer)
In this manuscript, Wang et al., present an interesting mechanism by which elevated TLR4 aggravates LPS-induced apoptosis in monocytes/macrophages from transgenic sheep. The methods are very well described and the results are presented clearly.
I would only have some questions and minor comments that would be worth clarifying in order to strengthen the manuscript:
What percentage of purity is achieved in the population of monocytes/macrophages stained with CD14 and CD11b? in supplementary figure 1 only immunofluorescence is shown, but the percentage of cells in the total population of PBMCs are actually monocytes/macrophages is not calculated.
What do you mean by dysfunctional autophagy? since this is only presented with figure 5a but is not strengthened by another method
In lines 146 and 266: remove "and"
Author Response
Thanks for your kind suggestions. We have provided a point-by-point response to your suggestions and re-submitted the revised manuscript. And all the changes in the revised manuscript are highlighted in the “Track Changes”.
Point 1. What percentage of purity is achieved in the population of monocytes/macrophages stained with CD14 and CD11b? in supplementary figure 1 only immunofluorescence is shown, but the percentage of cells in the total population of PBMCs are monocytes/macrophages is not calculated.
Response: We appreciate your professional advice and pay our homage to your kind work. We obtained PBMCs by gradient density centrifugation using a specific separation medium (Tbdscience) designed for sheep peripheral blood. Generally, PBMCs are 70-90% lymphocytes and 10-30% monocytes. Our experiment used only the adherent cells after 2h incubation. The non-adherent cells (which are almost lymphocytes) were washed out using PBS (see section 2.2 Animals and cells isolation, culture and identification). The specific markers of monocytes/macrophages, CD11b and CD14, were detected on the cell membranes by antibody staining to further identify these adherent cells. Figure S1 showed that nearly all these adherent cells were CD11b+/CD14+ (>90%, analyzed not fewer than 500 cells). These CD11b+/CD14+ cells mainly perform the function of removing pathogenic microorganisms. We, therefore, believe that it is feasible to use these cells as research subjects.
Point 2. What do you mean by dysfunctional autophagy? since this is only presented in Figure 5a but is not strengthened by another method
Response: A complete autophagy process consists of three crucial stages: autophagosome formation stage, autophagosome-lysosome fusion stage, and autolysosome degradation stage. In the process of autophagy, if autophagy is blocked at different stages, cells may fail to adapt to the stress, leading to apoptosis. By using transmission electron microscopy, we first found that the number of autophagosomes/autolysosomes in the TG group was significantly higher than in the WT group (Figure 4). The increase in the number of autophagosomes/autolysosomes may be due to a rise in autophagosome formation or a decrease in the level of autolysosome degradation. We further use Baf A1 to evaluate autophagic flux. Indeed, the Baf A1 treatment experiment confirmed that TLR4 overexpression increased the synthesis of autophagosomes (supplementary figure S2). p62 is a receptor protein related to the delivery of cargo. The decreased p62 protein and the increased LC3B-II/GAPDH ratio indicate the promoted autophagic flux. The increase of p62 accompanied by the rise of the LC3B-II/GAPDH ratio indicates the blockage of autophagic flux. The results shown in Figure 5 suggested that a high dose of LPS might block autophagic degradation in monocytes/macrophages of transgenic sheep. Moreover, the knockdown of TLR4 also promoted p62 degradation, which suggested that the blockage of autophagic flux was alleviated (Figure 7). Through a series of experiments, we have confirmed that the elevated TLR4 aggravates LPS-induced monocytes/macrophages apoptosis by leading to lysosomal dysfunction and impaired autophagic flux.
Point 3. In lines 146 and 266: remove "and"
Response: Thank you for your kind reminder. We have reformatted these words, please check our revised manuscript.
This manuscript is a resubmission of an earlier submission. The following is a list of the peer review reports and author responses from that submission.
Round 1
Reviewer 1 Report
The manuscript is quite interesting and well-designed, written, and discussed. but need some minor modifications as follows:
1. Line 107: references should be similar way.
2. Authors should provide the experiment conduct permission code no, provided by university ethical body.
3. Figure 2F: need non-breakable figure.
4. Figure 3C and F: need non-breakable figure.
5. Figure 5E: need non-breakable figure.
6. Figure 7D, H, and L: please provide non-breakable figure.
7. Figure 8B and D: please provide non-breakable figure.
8. Figure 9E and H: please provide non-breakable figure.
9. Figure 10: also need non-breakable figure.
Author Response
Response: We appreciate your professional advice and pay our homage to your kind work. To improve this manuscript, we first provide the experimental license code (see section 2.1 Animal ethical statement). And then, there is a clipping of all the figures on the right side, resulting in several graphs missing data. So we have reformatted these figures. Moreover, according to your suggestions, we re-edit the language of the article by native English speakers. We really hope that the flow and language level have been substantially improved. And all the changes in the revised manuscript are highlighted in the “Track Changes”. Please check the submitted revision manuscript.
Reviewer 2 Report
The manuscript by Wang et.al. describes a TLR4 transgenic Sheep model and proposes that TLR4 signalling in transgenic sheep triggers excessive activation of nfkb and p38MAPK signalling, along with modulation of autophagy and apoptosis. The paper focuses specifically on PBMC treatment with LPS. The authors propose that increased TLR4 signals in transgenic sheep cause altered autophagy and apoptotic responses. There are some problems with the paper as it is at the moment.
One major concern that I have is the extremely high amount of LPS used for the study. TLR4 is a very sensitive receptor and can typically be activated with low nanogram amonts of LPS. The authors have however , chosen to use from 1 to 100µg/mL of LPS . Indeed, some of the differences shown seem to rather only happen at 100µg/mL. It is unlikely that such an amount is ever going to be seen in vivo, begging the question, why have the authors chosen this dose? What happens at lower doses such as 100ng/mL?
Also the apoptosis assays are rather correlative. Simply showing that there is a difference in the ratio of Bax and BCL2 does not mean there is active apoptosis happening. There are multiple other pro and anti-apoptotic proteins at play too, that have overlapping functions. The authors should at the very least show caspase activity being different between the TG and wt. What is also not considered is a possible role for caspase-8 which can activate after TLR4 and can also lead to apoptosis.
As most of the paper is done using PBMCs it is important to have some information about what sort of cells are being used. This is however lacking. The methods section states that they used adherent cells from PBMC. This is likely then a mixture of macrophages, dendritic and neutrophils or some myeloid type cell, mixture. This should be defined in some more specific way. Which cells are dying in the apoptosis assays?
Overall I do not find the assays really support the conclusions and the work needs to be made more specific in terms of analysing apoptosis and autophagy. I have a significant question too about the purpose of making these sheep. Sheep already have a TLR4 system without it being overexpressed. Most of the questions being looked at here could be answered with a normal wild type sheep, especially if only PBMCs are being used for assays. How does using a transgenic TLR4 sheep tell us about normal TLR4 responses in larger mammals, which was the stated goal of the study.
Specific issues:
There is clipping of all the figures on the right side, resulting in several graphs missing data. This of course needs to be fixed.
The ethics statement is lacking any detail and needs to be updated in accordance to the author guidelines.
Figure 1A: there is very little information about how these animals were created. Was it a random insertion of the construct or was it targeted?. Do the auhtos know where the contruct has inserted? Is it in another gene?
Figure 2: it is not clearly stated what tissue this figure is using. Fomr the text it may be PBMC, but the legend should make this clear. Additionally, figure 2C does not really show much overexpression of TLR4. Do the authors have an explanantion for this relatively poor expression.
Figure 5A. The microscopy is not of high quality and it is difficult to make out any autophagosomes or see a clear difference between wild type and TG samples. It would be helpful to see some co-localisation of lc3 and Lysosomal markers if antibodies are available for sheep lamp2 for example.
Figure 5c shows LC3II/GAPDH ratio, but text refers to LC3II/LC3I ratio. Additionally, this ratio is not so reliable alone when analysing autophagy due to differences in the sensitivity of lc3 antibodies for lc3I and lc3II (see guidelines for use and interpretation of autophagy paper) and various other regulations. It would be preferable to also see some analysis of autophagic flux to make the claims that the authors are making.
Figure 5D, that authors note that 1µg/mL LPS reduced p62, wherease 100µg/mL increased it. They conclude that this means there is a blockage of autophagic degradation in the higher dose. P62 is also a known target of NFkB and this may just reflect the increased NFkB seen at the higher dose. Careful analysis of the half life of p62 in the presence or absence of chloroquine for example would be needed to confirm this observation as being due to changes in autophagic flux.
Author Response
Thanks for your kind suggestions. We have provided a point-by-point response to your suggestions and re-submitted the revised manuscript. Moreover, the figures have been reformatted. In addition, according to your suggestions, we re-edit the language of the article by native English speakers. We really hope that the flow and language level have been substantially improved. Please see the attachment and check the submitted revision manuscript

Reviewer 3 Report
This study aimed at identifying a relation between the TLR4 induced processes of apoptosis and autophagy is potentially very interesting. However my main concern is that all experiments are conducted using total PBMCs. To be able to show such relationship, experiments should be conducted with TLR4-expressing cells only, or at least it should be clear in what cell types apoptosis/autophagy occur and if these cells express TLR4.
Also: not sure if this could also be done using human or murine cells, why generate TLR4 transgenic sheep? The relevance of using sheep to assess the proposed interaction between TLR4 induced processes of apoptosis and autophagy should be better substantiated
Not shown which cell subsets express Tg TLR4; this is important regarding the comment on the use of total PBMCs rather than purified TLR4 expressing cells or identifying cell subsets
Author Response
Thanks for your kindly suggestions. We have provided a point-by-point response to your suggestions and re-submitted the revised manuscript. And all the changes in the revised manuscript are highlighted in the “Track Changes”.
Point 1. However my main concern is that all experiments are conducted using total PBMCs. To be able to show such a relationship, experiments should be conducted with TLR4-expressing cells only, or at least it should be clear in what cell types apoptosis/autophagy occurs and if these cells express TLR4. Not shown which cell subsets express Tg TLR4; this is important regarding the comment on the use of total PBMCs rather than purified TLR4 expressing cells or identifying cell subsets
Response: We appreciate your professional advice and pay our homage to your kind work. Actually, we have already confirmed the expression level of TLR4 in PBMCs (We have re-titled the Figure2, please check our revised manuscript). The mRNA and protein levels of TLR4 in PBMCs have been shown in Fig 2A to C. And PBMCs from both wild-type sheep and transgenic sheep express TLR4. By immunofluorescence and flow cytometry analysis, it was also shown in other papers of our research group that almost all PBMCs express TLR4 [1,2]. Moreover, our previous reports have already identified these cells [1,3]. The specific markers of monocytes/macrophages CD14, CD11b and F4/80 were detected on the cell membranes by staining. These cells mainly perform the function of removing pathogenic microorganisms. We, therefore, believe that it is feasible to use PBMCs as a research subject.
Point 2. Also: not sure if this could also be done using human or murine cells, why generate TLR4 transgenic sheep? The relevance of using sheep to assess the proposed interaction between TLR4-induced processes of apoptosis and autophagy should be better substantiated
Response: Many research groups have used mouse models of TLR4 mutations to explore pathological changes in living organisms [4,5]. These studies suggest that TLR4 plays an important role in processes such as inflammation, immune and cell death. However, few studies have reported the use of TLR4 mutations in large mammals. Some studies suggest that the inflammatory response elicited by TLR4 may enhance the host's ability to eliminate pathogenic microorganisms, whereas other investigations have reported that TLR4-mediated responses may inflict harm upon the organism, contingent upon the experimental models employed and the in vivo and in vitro conditions. Therefore, further research is warranted to unravel the underlying molecular mechanisms. Here, we present a novel transgenic mammalian model characterized by an enriched expression of TLR4 and further explore its impact on autophagy and apoptosis. Sheep are one of the most important livestock resources and are considered the more suitable animal model for biomedical research because they share similar physiological and genetic characteristics with human beings. Furthermore, many diseases caused by Gram-negative bacterial infections (such as Brucella, Salmonella and E. coli, which are mainly recognized by TLR4) have seriously affected the sheep industry. Therefore, the preparation of a TLR4 mutant transgenic sheep model may be of interest to animal geneticists studying molecular breeding for infection resistance and individuals interested in bacterial infections. And the TLR4-overexpressing sheep also serve as a good large mammal model for disease resistance breeding.
In addition, the figures have been reformatted. Moreover, according to your suggestions, we re-edit the language of the article by native English speakers. We really hope that the flow and language level have been substantially improved. Please check the submitted revision manuscript.
- Li, Y.; Zhao, Y.; Xu, X.; Zhang, R.; Zhang, J.; Zhang, X.; Li, Y.; Deng, S.; Lian, Z. Overexpression of Toll-like receptor 4 contributes to the internalization and elimination of Escherichia coli in sheep by enhancing caveolae-dependent endocytosis. J Anim Sci Biotechnol 2021, 12, 63, doi:10.1186/s40104-021-00585-z.
- Wang, M.; Qi, Y.; Cao, Y.; Zhang, X.; Wang, Y.; Liu, Q.; Zhang, J.; Zhou, G.; Ai, Y.; Wei, S.; et al. Domain fusion TLR2-4 enhances the autophagy-dependent clearance of Staphylococcus aureus in the genetic engineering goat. Elife 2022, 11, doi:10.7554/eLife.78044.
- Wei, S.; Yang, D.; Yang, J.; Zhang, X.; Zhang, J.; Fu, J.; Zhou, G.; Liu, H.; Lian, Z.; Han, H. Overexpression of Toll-like receptor 4 enhances LPS-induced inflammatory response and inhibits Salmonella Typhimurium growth in ovine macrophages. Eur J Cell Biol 2019, 98, 36-50, doi:10.1016/j.ejcb.2018.11.004.
- Chen, S.N.; Tan, Y.; Xiao, X.C.; Li, Q.; Wu, Q.; Peng, Y.Y.; Ren, J.; Dong, M.L. Deletion of TLR4 attenuates lipopolysaccharide-induced acute liver injury by inhibiting inflammation and apoptosis. Acta Pharmacol Sin 2021, 42, 1610-1619, doi:10.1038/s41401-020-00597-x.
- Bhattarai, S.; Li, Q.; Ding, J.; Liang, F.; Gusev, E.; Lapohos, O.; Fonseca, G.J.; Kaufmann, E.; Divangahi, M.; Petrof, B.J. TLR4 is a regulator of trained immunity in a murine model of Duchenne muscular dystrophy. Nat Commun 2022, 13, 879, doi:10.1038/s41467-022-28531-1.
Round 2
Reviewer 2 Report
I have read through the authors response and the revised manuscript, but am sorry to say say that I feel that the manuscript is still not ready for publication. While the authors have corrected their manuscript in response to some of my comments, others have not really been adequately addressed. for example I had asked them to define the cells that they are using in more detail, but they have just said that they have done this in other papers. This is not enough. I also have a major concern about how much weight is being put into the p62/atg5/LC3II data to suggest that autophagic flux is blocked. I asked the authors to do some flux analysis, but they have not done this. Given how major this claim is to the manuscript It is still not adequately explained experimentally.
Additionally I do not feel the paper really explains itself in a way that there is a convincing story here. Essentially the authors show that when they overexpress tlr4 there is a major increase in signals that are downstream of TLR4 when there is a very high dose of LPS used. they claim that this is bad because it blocks autophagy which leads to accumulation of reactive oxygen species and overexpression of cytokines like TNF and IL1b, which seems to be dependent on the blocked autophagy. It could also all juts simply be due to having much higher signals form TLR4 which you would expect, leading to higher transcription factor activation and so on. From the data the authos have rather shown that overexpressing a LTR leads to more activation of the signals downstream of that TLR and if you block those signals with siRNA or drugs it reverses the effect of overexpression. This is not particularly informative mechanistically. To block authophagy and reverse the effects of tlr4 over expression the authors have used 3MA, which almost certainly blocks many other pathways, not specifically autophagy, being a general pi3k inhibitor.
the main point of the paper was to claim that high doses of TLR4 block autophagy in TLR4 transgenic pbmcs and that this contributes to worse outcomes for transgenic sheep in highly concentrated infection models. The basic point of autophagy blockage is however not really confirmed from the data and the rest of the story is rather confirmation that the TLR4 over expression is more active than without it.
the Apoptosis section is improved by addition of active caspase-3 blots, but still not really explained in terms of understanding why tlr4 overexpression causes this. I had asked the authors to also reference at least caspase-8, but this has also not been done. Perhaps there are no caspase-8 antibodies for sheep, but to completely ignore it is not helpful.
Overall I cannot accept the paper as these above issues are still not clearly resolved.
Reviewer 3 Report
the manuscript has been improved by incorporating suggestions from reviews. however still two concerns remain:
1. It is still unclear which cells are being used in experiments. The authors talk about "PBMCs". PBMCs comprise lymphocytes (T/B/NK cells), monocytes, dendritic cells and a small fraction of granulocytes.
In the methods there is 1 sentence mentioning: pbmcs are cultured for 2h at 37C, and the non-adherent cells are washed off. Thus labeling the cells used in experiments as "PBMCs" is not right. What is the composition of the adherent cells that are cultured? Besides monocytes, what other cells are present, and do they express TLR4 as well?
The authors should provide flow cytometry data to show the composition of the cell prep used in experiments.
accordingly labeling of all figure headings/text should be adapted from "PBMCs" to monocytes or monocyte/macrophage
2. the high dosis of LPS (100 ug/mL) required to obtain certain differences is not very physiological.
3. the use of TLR4 Tg sheep to assess the role of TLR4 in autophagy and apoptosis is still not substantiated